# Glycosylation Regulation by TMEM230 in Aging and Autoimmunity

**DOI:** 10.3390/ijms26062412

**Published:** 2025-03-07

**Authors:** Eleonora Piscitelli, Edoardo Abeni, Cristiana Balbino, Elena Angeli, Cinzia Cocola, Paride Pelucchi, Mira Palizban, Alberto Diaspro, Martin Götte, Ileana Zucchi, Rolland A. Reinbold

**Affiliations:** 1Institute for Biomedical Technologies, National Research Council, 20054 Segrate, Italy; eleonora.piscitelli@itb.cnr.it (E.P.); edoardo.abeni@gmail.com (E.A.); cinzia.cocola@itb.cnr.it (C.C.); paride.pelucchi@itb.cnr.it (P.P.); 2I.R.C.C.S. Ospedale Galeazzi-Sant Ambrogio, 20157 Milan, Italy; balbino.cristiana@yahoo.it; 3Department of Physics, University of Genoa, 16146 Genoa, Italy; elena.angeli@unige.it (E.A.); alberto.diaspro@iit.it (A.D.); 4Department of Gynecology and Obstetrics, University Hospital of Münster, 48149 Münster, Germanymartin.goette@ukmuenster.de (M.G.); 5Nanoscopy, Istituto Italiano Tecnologia, 16152 Genoa, Italy; 6Associazione Fondazione Renato Dulbecco, Via Fantoli 16/15, 20138 Milan, Italy

**Keywords:** transcriptomic single cell sequencing, aging, autoimmunity rheumatoid arthritis, RNASET2, TMEM230 (C20orf30), glycosylation, endoplasmic reticulum unfolded protein response, PI3K-AKT-mTOR signaling, Parkinson’s and Alzheimer’s disease

## Abstract

Aging is often a choice between developing cancer or autoimmune disorders, often due in part to loss of self-tolerance or loss of immunological recognition of rogue-acting tumor cells. Self-tolerance and cell recognition by the immune system are processes very much dependent on the specific signatures of glycans and glycosylated factors present on the cell plasma membrane or in the stromal components of tissue. Glycosylated factors are generated in nearly innumerable variations in nature, allowing for the immensely diverse role of these factors in aging and flexibility necessary for cellular interactions in tissue functionality. In previous studies, we showed that differential expression of TMEM230, an endoplasmic reticulum (ER) protein was associated with specific signatures of enzymes regulating glycan synthesis and processing and glycosylation in rheumatoid arthritis synovial tissue using single-cell transcript sequencing. In this current study, we characterize the genes and pathways co-modulated in all cell types of the synovial tissue with the enzymes regulating glycan synthesis and processing, as well as glycosylation. Genes and biological and molecular pathways associated with hallmarks of aging were in mitochondria-dependent oxidative phosphorylation and reactive oxygen species synthesis, ER-dependent stress and unfolded protein response, DNA repair (UV response and P53 signaling pathways), and senescence, glycolysis and apoptosis regulation through PI3K-AKT-mTOR signaling have been shown to play important roles in aging or neurodegeneration (such as Parkinson’s and Alzheimer’s disease). We propose that the downregulation of TMEM230 and RNASET2 may represent a paradigm for the study of age-dependent autoimmune disorders due to their role in regulating glycosylation, unfolded protein response, and PI3K-AKT-mTOR signaling.

## 1. Introduction

Aging is thought to be associated with loss or dysregulation of the function of the endomembrane system (ES) of cells [1,2,3]. The ES includes membrane-bound organelles such as the endoplasmic reticulum (ER), Golgi complex and lysosomes and the system of intracellular cytoskeletons. The intracellular cytoskeleton system is a complex and constantly changing network of interlinking glycoprotein intracellular filaments/extracellular scaffolds that determines cell fate and function of the tissue. This network regulates cell-to-cell and cell-to-stromal interactions, adhesion and signaling [4,5]. The specificity of interactions is largely based on glycan sequence and structure that are covalently attached to intracellular and extracellular “glycosylated” factors. These include proteoglycans covalently linked to glycosaminoglycans (GAG), glycoproteins or glycolipids. Interactions can be direct and constant or dynamic and long-distance through intracellular trafficking or secretion of glycosylated factors utilizing the cytoskeleton system or stromal scaffolds. Many of the dynamic interactions of intracellular filaments are ATP-driven and dependent on the oxidative phosphorylation metabolic pathway. ATP is also essential for motor protein (such as kinesins) dependent glycan-containing cargo trafficking, such as kinesins. The glycan-containing cargo is destined to the plasma membrane, all organelles and non-membrane bound bodies, or secreted to the extracellular environment of cells [6,7,8,9]. Motor proteins are powered by ATP generated by mitochondria through oxidative phosphorylation. The trafficking of cargo in cells and tissue is necessary for the regeneration of the membrane of organelles or plasma membrane of cells for homeostasis, aging or normal cellular repair following injuries or insults. Additionally, trafficking is necessary for the turnover of membrane components to allow for dynamic cell signaling, interaction, modification, and modulation of the extracellular environment. In contrast, loss of or aberrant regulation of the ES results in disease development such as fibrosis, cancer and autoimmunity [4,10]. Trafficked and secreted components are generally glycoconjugates due to the nearly innumerable variations in glycan structure and 3-dimensional (3D) chain formation capacities that exist in nature. This provides the vast flexibility necessary for diverse and unique molecular interactions of cells in tissue. Likely, the linkage of proteins or lipids by ER-generated glycans (carbohydrates) was essential for the evolutionary development of different types of tissue in metazoans and host immunity against pathogens. Previously, we showed that TMEM230 (a membrane-bound protein of the ER) and RNASET2 (a lumen component of lysosomes) function together in intracellular trafficking and turnover of damaged or non-essential intracellular components [6,7,8,11,12,13,14]. Lysosome and proteasome complex-dependent recycling of cellular components are necessary for regeneration (following injury or disease) of organelle and plasma membrane components to prevent intracellular buildup of toxic or cell-damaging aggregate forming proteins or amyloid structures. Concurrent with our studies, specific human gene mutations were discovered, which may drive aberrant TMEM230 protein structure formation in Parkinson’s disease (PD) and Alzheimer’s disease (AD) [15,16,17]. Gene mutations identified in PD and AD suggest that TMEM230, as an integral membrane protein of the ER protein, may contribute to ER-dependent protein stability or quality control mechanisms in the synthesis of proteins, such as in Lewy bodies [18,19,20,21]. Additionally, the ER regulates organelle trafficking, including mitochondria and therefore has a role in mitochondrial-associated metabolic activities, such as localizing the mitochondria to strategic subcellular regions of the cell [16]. Protein stability or quality control steps are initiated in the ER [7,8,9,16,22,23,24,25,26,27,28,29]. Mutated forms of TMEM230 transcripts may, therefore, stimulate neural tissue cell death by autophagy of dopaminergic (DA) neurons by mitochondria stress or cytotoxic protein aggregation in the ER. Proteins destined for the plasma membrane or the mitochondrial membrane (such as glycoproteins) or secretion (proteoglycans) are generated by ribosomes that are docked to the rough ER. As the proteins are synthesized, they are simultaneously translocated into the ER lumen, where they are then glycosylated and undergo proper protein folding by diverse chaperones [30,31,32,33,34,35,36,37,38]. Misfolded proteins are identified by quality control factors, including heat shock proteins and retrotranslocation by ER-associated degradation (ERAD) via proteasome complexes for proteolysis [34,35]. Recognition of unfolded or misfolded proteins depends in part on the glycans that are conjugated to the proteins that promote the detection of aberrant protein substructures.

While mutated forms of TMEM230 gene products play a pivotal role in the development of many neurodegenerative disorders, such as PD and AD, likely due to the accumulation of misfolded or unfolded proteins, our recent studies strongly support that aberrant regulation of expression of non-mutated forms of TMEM230 may also promote other human diseases or disorders. TMEM230 has polytropic interactions in diverse cell types and human tissues through its role in immune system activities such as cell-to-cell recognition, cell-to-substratum interactions, antigen processing and presentation and glycosylation of antibodies that occur through the endomembrane system (ES) and are initiated in the ER. Loss of normal ES activities in cancer and autoimmune disease development are well established [32,39,40,41,42,43]. In multicellular organisms, cancer, in part, is due to tumor cells that fail to maintain contact inhibition within tissue boundaries demarcated by glycoconjugates of different cell types and extracellular scaffolds. Oncogenesis may also be initiated when an organism fails to recognize immunologically rogue-acting tumor cells that have developed aberrant glycan cell surface signatures [44,45,46,47,48]. Many genes responsible for the establishment of cancer or autoimmune disorders are misregulated genes that control intracellular cargo trafficking or glycosylation in the ES [4,7]. The commonality of these human pathologies is that they are age-dependent human diseases that utilize glycoconjugates to maintain proper glycan-dependent cell-to-cell, cell-to-matrix substrate, and immune system interactions. Nevertheless, the exact causes of autoimmune diseases remain largely unknown. However, research has suggested that a combination of genetic and environmental factors, as well as certain pathogens, may contribute to the development of human autoimmune disorders. In a healthy immune response, self-reactive cells are generally destroyed and eliminated before they become active, made unreactive by energy, or suppressed by other regulatory cells. Immune tolerance prevents the immune system from attacking the body’s own cells. When this process fails (T and B cells react with self-proteins), such as in age-dependent loss of differentiation potential or depletion of specific types of immune system cells, the immune system may produce antibodies against its own tissues, leading to an autoimmune response. We previously have shown that TMEM230, an ER-associated protein, regulated ATP-powered motor protein trafficking of intracellular cargo and secretion [7,8]. We also demonstrated that misexpression of the levels of TMEM230 and RNASET2 contributed to aggressive cancer features, including aberrant angiogenesis and destructive tissue remodeling, and a loss of normal cell-to-cell and cell-to-substratum attachments. These aggressive features are also hallmarks of rheumatoid arthritis, an autoimmune disorder that occurs with aging. The reason why misexpression of the levels of TMEM230 and RNASET2 occurs in certain cancers is unknown to us. Recently, we identified that TMEM230 and RNASET2 as downregulated in RA compared to OA patients, suggesting that their misregulation of expression was due to age dependency, as RA is considered an age-dependent disorder compared to OA [7].

In this study, by comparing single-cell sequencing transcription profiles of samples obtained from patients with RA or OA, we identified genes and pathways co-regulated with TMEM230 and RNASET2 that distinguish RA from OA cells. We found that in RA patient cells, TMEM230 and RNASET2 may have diverse roles due to being essential proteins of the ES. Downregulation of TMEM230 and RNASET2 was observed in cell types of the diseased tissue of RA patients, including the immune system, blood vessels and synovial fibroblast cells, compared to OA. The genes and pathways identified were associated with oxidative phosphorylation and reactive oxygen species pathway (ATP synthesis by mitochondria) and ER-dependent functions such as xenobiotic metabolism, unfolded protein response, glycosylation, hypoxia, inflammation, and apoptosis. While mitochondria are often not considered part of ES trafficking, ES associated vesicular cargo trafficking is dependent largely on ATP to power motor proteins. The ER is the initial organelle where glycosylated factors such as proteins are synthesized, structurally modified and evaluated in terms of quality control. Our new study suggests that further investigations are justified in evaluating whether TMEM230 and RNASET2 may have roles in aberrant immune system regulation and destructive tissue remodeling associated with aging. In particular, TMEM230 and RNASET2 expression were absent or very low in all cell types, including immune system cells such as T and B cells, professional antigen-presenting and phagocytic cells (macrophage, dendritic cell, or synovial fibroblast) our analysis suggested that loss of expression of these genes are novel markers that correlate with loss of normal endomembrane activities in trafficking and glycosylation in autoimmunity or aging.

### Background

We previously identified *tmem230*, as a transmembrane protein of the endoplasmic reticulum (ER) and a master regulator of angiogenesis in early zebrafish development [11]. Modulation of Tmem230 expression by itself was sufficient to rescue an improper number of endothelial cells (ECs) induced by aberrant expression or inhibition of the activity of genes associated with the Notch receptor and ligand pathway [11]. In humans, the Notch receptors and ligands are a family of glycosylated transmembrane proteins that mediate direct cell-cell interactions and control cell-fate specification. How TMEM230 modulates vessel-network formation and glycosylation in early vascular development, wound healing and disease is a focus of our continuing research. Loss of normal vessel network formation and function are hallmarks of many age-dependent autoimmune disorders [7,8,11,12,13,14]. We previously identified that patients with age-associated disorders such as giant cell arteritis expressed lower levels of TMEM230 in blood vessels and in peripheral blood immune cells [7,8,11,12,13,14]. The ER mediates the earliest processes in glycan synthesis and glycan conjugation to proteins, lipids and carbohydrates [33,36,49]. Glycans have diverse roles, including proper folding, stability, and solubility of proteins and therefore are essential in intracellular trafficking of cargo to organelles and the plasma membrane and in secretion of cell factors [8,49,50]. ER and Golgi Apparatus-dependent trafficking of glycoconjugates are essential in the maintenance and repair of cell and organelle membranes in aging. Concurrent with our TMEM230-associated studies, specific human gene mutations were identified promoting aberrant TMEM230 protein structure formation in age-dependent neurodegeneration disorders, including Parkinson’s disease (PD) and Alzheimer’s disease (AD) [15,16,17]. Misfolded proteins are identified in cells by quality control processes by ER-associated degradation (ERAD) [34,35]. Loss of ERAD activity may promote amyloid formation, resulting in neurological pathologies. As a result, the ER and, hence, TMEM230 play vital roles in the development of diseases of many age-dependent neurodegenerative disorders [16]. Our long-term research focus is to identify whether loss of TMEM230 expression is also associated with other human autoimmunity-based disorders.

## 2. Results

### 2.1. Dysregulated Genes in Autoimmunity

#### Single Cell RNA Sequencing Data Analysis

Single-cell RNA sequencing data from publicly available datasets of synovial tissue were acquired from the Gene Expression Omnibus repository (GSE152805) for OA and (GSE200815) for RA patients [7]. Single-cell transcriptomic analysis was performed with 4 RA and 3 OA datasets (Chromium 10× Genomics). Integration of the OA and RA datasets was performed to allow the combination of the different cell clusters of synovial tissue to be compared between RA and OA. Essentially, corresponding filtered feature barcode matrix files were imported in R (4.1.0) and analyzed by the Seurat package (V.4.0.6) as previously described [13]. The different clusters of patient cells were defined by specific markers from the literature, shown in Figure 1 and previously described [6]: (SF) synovial fibroblast 1, (Endo1) endothelial, (POSTN_SF) synovial fibroblast defined by POSTN gene, PRDG4_SF (synovial fibroblast defined by PRG4 gene), IS_B-T-NK (immune system B, T, Natural Killer cells), (CXCL12_SF) synovial fibroblast defined by CXCL12 gene, (MP) macrophages, (SM) smooth muscle (predominantly from blood vessels), (DC) dendritic cells (macrophage like), (Endo2) endothelial cells (blood vessel), (SF2) synovial fibroblast 2, (PDC) plasmacytoid dendritic cells, (Endo3) endothelial cells (blood vessels), and (IS_O) immune system cells “Others” [51].

Our previous studies supported that TMEM230 generally regulated ER-localized enzymes involved in glycan synthesis, processing, modulation and glycosylation, such as glycosyl transferases and glycosyl hydrolases (Table 1) [6,7,8,11,12,13,14].

ER scaffold protein TMEM230 and lysosomal enzyme RNASET2 expression were detected in the majority of cell-type clusters (Figure 2 and Figure 3). In support that TMEM230 may have a role in autoimmunity, TMEM230 was detected as downregulated in synovial tissue cell types from patients with RA compared to OA by single cell transcriptomic sequencing analysis ). Down-regulation of TMEM230 was prominent in blood vessel endothelial cells (Endo 1 and Endo 2) and smooth muscle (SM) cells that surround the endothelial cells. We previously characterized the role of TMEM230 in blood vessels as being necessary to form functional blood vessels. When TMEM230 was aberrantly overexpressed, aggressive blood vessels were generated that were “leaky” due to loss of normal endothelial and smooth muscle cell] contacts [7,12,13]. These types of aberrant blood vessel features were associated with intravasation and extravasation of immune cells, as we previously observed [7,12,13]. Homing intravasation and extravasation capacity of immune cells into blood vessels regulated by TMEM230 was supported by loss of detectable levels of TMEM230 expression in immune system T, B and natural killer cells (IS_BTNK, Figure 2 and Figure 3). TMEM230 was also downregulated in all synovial fibroblast cells (PRG4_SF, POSTN_SF, SF1, SF2, and CXCL12_SF). Essentially, TMEM230 was downregulated in all cell types of the diseased synovial tissue of RA patients, suggesting that “proper” levels of TMEM230 is necessary in maintaining or repairing joint tissue in young patients. OA patients were considered as “healthy” controls in this study as they were not afflicted with with chronic inflammation and the pathological features associated with autoimmunity. We previously demonstrated that specific levels of TMEM230 expression were necessary for normal phagocytic activity of macrophage, fibroblast and glial cells [7,12,13]. Sustained down- or up-regulation of TMEM230 aberrant expression promoted chronic destructive tissue remodeling by phagocytic cells [7,12,13]. Additionally, RNASET2, a lumen component of lysosomes, was previously identified by our group to have a role in the immune system through its role in the recycling of cellular debris and components following injury, disease or infection [7,8,9]. The function of RNASET2 in stress, infection or injury is necessary to prevent pathogen cell replication and intracellular buildup of toxic or cell-damaging “free” RNA. Additionally, immune system activities such as glycosylation or turnover of antibodies or processing of pathogenic RNA antigens occur through lysosome activities regulated in part by RNASET2 and the endomembrane system [6]. Loss of normal endomembrane system activities that contribute to cancer and autoimmune disease development are well established [32,39,40,41,42,43]. As for TMEM230, we observed downregulation of RNASET2 in RA patient cells compared to OA (Figure 3). RNASET2 displayed decreased expression in immune cells (predominantly in dendritic, DC and “other” immune system cells, IS-O) and synovial fibroblasts (SF1, CXCL12, and SF2) in RA. RNASET2 expression was also downregulated in blood vessel Endo3 endothelial cells.

To determine which genes may be modulated in the different cell types of the synovial tissue due to downregulation of TMEM230 and RNASET2, we compared gene expression profiles for the different cell clusters in RA and OA patients (Appendix A). The molecular pathways identified supported the roles of TMEM230 and RNASET2 as being integral to the ER and lysosome activities, respectively. All the pathways identified in this study were previously linked with the hallmarks of aging and physiological and cellular stress. In particular, the pathways indicated change in the regulation of mitochondria-dependent oxidative phosphorylation and reactive oxygen species, ER-dependent stress and unfolded protein response, DNA stress and repair (UV response and P53 signaling pathways), cell stress (hypoxia and epithelial-to-mesenchyme transition), and senescence, glycolysis, metabolism and apoptosis regulation through PI3K-AKT-mTOR signaling, coagulation, immune system response to cytokines (interferon and IL6) (Figure 4, Figure 5, Figure 6 and Figure 7). Change in oxidative phosphorylation correlated with our previous observation that cytoskeleton motor activity powered by mitochondria-generated ATP was necessary for ES-dependent intracellular trafficking and secretion and regulation of cell morphology through regulation of the intracellular cytoskeleton system [6,7].

Phalloidin was used to investigate the distribution of filament protein F-actin in RA patient cells in which TMEM230 expression levels were modulated. Phalloidin prevents ATP-induced depolymerization of F-actin fibers and specifically interacts at the interface between F-actin subunits. Increasing expression of TMEM230 recapitulates morphologically actin-based growth cone motility and guidance as commonly associated with cell extensions that facilitate growth cones, which is especially evident in cells associated with normal or tumor neural tissue where actin filaments are extended by ATP-dependent polymerization into longer “filamentous” threads. By comparing U87-MG (cells that recapitulate glial cell tumor and morphological properties of aggressive gliomas of patients with glioblastoma multiforme) control and cells in which levels of TMEM230 were upregulated (Figure 8A,B), we observed that control cells appear with a more “globular” morphology associated with monomer units of actin. Similarly, globular morphology associated with monomer units of actin was also observed using synovial fibroblast cells derived from patients with RA and OA when TMEM230 or RNASET2 expression was modulated independently (Appendix A). This supports the fact that the linkage of proteins or lipids by ER-generated glycans has a role in active and dynamic cytoskeleton network remodeling and cell and tissue morphology in age-dependent autoimmunity.

Our results supported TMEM230 may represent a key target for therapeutic intervention in other autoimmunity disorders and cancers through its role in regulating the expression of enzymes in glycan synthesis and processing and glycolysis. As the ER regulates protein stress response, not surprisingly, we also identified genes in ER pathways, such as unfolded protein stress response, that may contribute to endomembrane-based disorders such as aging and autoimmunity. Recognition of misfolded proteins depends in part on the glycans that are conjugated to the proteins that promote the detection of substructures within proteins, such as exposed hydrophobic regions or immature glycans (Figure 9). Collectively, our study supports that TMEM230 may modulate glycosylation, proteoglycan and glycosaminoglycan synthesis for maintaining tissue homeostasis, repair or regeneration. As RA is often associated with aging, we may generalize that proper levels of TMEM230 may be necessary for maintaining optimal joint tissue homeostasis, repair or regeneration in aging.

In support of the candidate role of TMEM230 in aging, mutated forms of the TMEM230 gene were previously identified in the development of many neurodegenerative disorders associated with aging, including Alzheimer’s and Parkinson’s diseases [15,16,17,18,19,20,21,22,23,24]. This connection of TMEM230 to neuodegneration supports the role of glycans in ER-dependent protein folding and stability and stress response is important in protecting or inducing unfavorable physiological conditions that develop with aging and may support additional roles of TMEM230 in the aging process. Accumulation of misfolded or unfolded proteins in the ER through loss of normal TMEM230 expression in aging may represent an important model of study in aging (Figure 10).

## 3. Discussion

The ER is an organelle that integrates endomembrane trafficking and signaling to sense and respond to extracellular environmental changes in tissue. Responses include promoting cellular processes for stress, inflammation and immune system activation. Loss of normal cell-to-cell and cell-to-substratum recognition and interactions results in loss of immune cell interactions within the tissue, blood vessels and extracellular matrix. Our previous studies supported the idea that TMEM230 modulated cellular processes with ER and extracellular matrix functions. These included promoting intracellular trafficking and secretion of cell factors and vesicles, homing and migration of cells, and tissue remodeling associated with normal wound healing or destructive tissue remodeling. These cell features are associated with human inflammation or autoimmune disorders and cancer. In previous studies, we showed that high levels of TMEM230 were associated with increased mitochondrial activities such as electron transport in oxidative phosphorylation and in upregulation of ATP synthesis that power motor protein trafficking of cargos on intracellular scaffolds [13]. The oxidative phosphorylation enzymatic pathway was identified as modulated with the downregulation of TMEM230 in RA patient cells in this study. Additionally, we also previously showed that change in oxidative phosphorylation with TMEM230 expression was coupled with a loss of synthesis of metalloproteins and in stress response cellular processes. This suggests that TMEM230 may have a central role in the metabolization of toxic compounds in autoimmunity or aging. TMEM230, as a regulator of metabolization of toxic compounds or drugs, has important implications for promoting or inhibiting autoimmune disorders, where a toxic environment may have an input in increasing the risk factor in pathogenesis. Concurrent with our early studies of *tmem230* regulation in an animal model, gene mutations in human patients were identified, suggesting a high risk of age-dependent neurological disorders, including Parkinson’s disease (PD) and Alzheimer’s disease (AD) development. The role of TMEM230 in autoimmunity is also supported by our previous study of GCA, a disorder of immune system dysfunction often observed in older patients. As an apparent regulator of the ER, TMEM230 may function in protein stress response as indicated by the genes differentially modulated in RA compared to OA. ER stress is associated with the accumulation of misfolded proteins in the ER lumen by the activation of unfolded protein response (UPR) pathways [8,30,31,32,33,37,49,50,52,53,54]. If chronically activated, UPR promotes cell death via apoptosis and mitochondrial dysfunction. Our sequencing analysis presented in this study supports that aberrant modulation of TMEM230 expression may promote ER-induced stress, cell death, and mitochondrial dysfunction in both cancer and RA (Appendix A).

Candidate roles of TMEM230 in ER regulation in RA patients is complex to interpret. We previously showed overexpression of TMEM230 promoted angiogenesis and destructive tissue remodeling by promoting aberrant “leaky” blood vessel formation in tumors [7]. Additionaly destructive tissue remodeling was also induced with upregulation of TMEM230 in U87-MG cells that recapitulate agressive glial phagocytic cell features. Tumor glial cells expressing TMEM230 displayed increased migration capacity and when confronted with ECs in their path in co-culture assays, infiltrated, enveloped or displaced ECs suggestive of the intussusceptive structural remodeling of blood vessels. Our previous gene expression analysis also uncovered aberrantly low levels of TMEM230 expression was correlated with loss of cell adhesion to extracellular substratum, supporting that loss of TMEM230 expression may also promote destruction of blood vessels by inability of blood vessels to integrate and adhere to their stromal environmen [7,8]. In this study, our evidence supports that misregulation of TMEM230 promotes abnormal glycosylation and therefore aberrant glycosylation may represent hallmark of RA and aging.

The synovial membrane, a connective tissue that protects the inner surface of the capsule of synovial joints and bursa, has direct interaction with the fibrous membrane on synovial fluid [7,42,55,56,57,58,59,60,61,62,63,64]. The synovial fluid at the tissue surface contains macrophage-like cells (MLC) and cells with fibroblast-like synoviocyte cells (FLS). The MCL cells maintain the synovial fluid and phagocytose debris from wear and injury. FLS cells produce hyaluronan, an extracellular glycosaminoglycan in synovial fluid. In this study, changes between RA and OA were observed primarily in cells with fibroblast and less so with cells with macrophage features cells (Appendix A). These results may provide greater insight into the development of pharmacological agents specifically targeting synovial cells with fibroblast phagocytic activities.

While we have manipulated the expression of TMEM230 using human-derived cell lines and primary blood vessel-forming endothelial cells, as our conclusions and sequencing analysis are derived from using patient whole tissue (RA, OA), further research will need to be performed in manipulating TMEM230 expression using patient-derived cells to generate 3D organoids and co-cultures assays that we have previously established to validate our expression profiles and conclusions [13].

## 4. Materials and Methods

Gene expression in RA versus OA clusters generated by single-cell RNA seq analysis.

The different cell populations from OA and RA were analyzed by comparing the different cell clusters, one by one, using Gene Set Enrichment Analysis (GSEA), as previously described [13]. To determine whether the genes were statistically differentially expressed in OA versus RA cell populations, the Kolmogorov Smirnov (K-S) test was used, considering statistically significant differences with a *p*-value of 0.05 or less, as previously described [13].

### 4.1. Cloning of Lentiviral-System-Based Construct for Modulating TMEM230 Protein Expression

The TMEM230 mRNA sequence (for upregulating TMEM230) or shTMEM230 (for upregulating native TMEM230 expression) was cloned into pcDNATM6.2-GW/EmGFP using the BLOCK-iTTM Pol II miR RNAi Expression Vector Kit with EmGFP (K493600, Thermo Fisher Scientific, Waltham, MA, USA) following the manufacturer’s instructions and as previously described [13]. Lentivirus particles were produced in HEK293T cells by transfecting pCDH or pLENTI vectors together with psPAX2 and pMD2.G (Addgene plasmids #12260 and #12259, gift from Didier Trono, Milan, Italy) as helper vectors for 2nd-generation viral packaging (with a ratio 4:3:1, respectively) using the LipofectamineTM 2000 Transfection Reagent (11668027, Thermo Fisher Scientific, Waltham, MA, USA) following manufacturer’s instructions and as previously described [13].

### 4.2. Adherent Cell Cultures

The human cells from RA and OA patients were obtained from informed patients affected by rheumatoid arthritis and osteoarthritis collected at the Ospedale Galeazzi, Sant Ambrogio at MIND (Milan, Italy). The approval of the Clinical Ethical Committee (Project Diamant) was released on 5 April 2023 by the Ethics Committee of the San Raffaele Hospital—Milan, with EC Opinion Register Number 104/2023. The human brain glioblastoma U87-MG cell line was obtained from the American Type Culture Collection (ATTC, Manassas, VA, USA). Human cells were isolated from synovial tissues and maintained in Dulbecco’s Modified Eagle’s Medium (DMEM, Euroclone, ECB7501L; Pero, Milan, Italy) supplemented with 10% fetal bovine serum (FBS, F7524; Sigma, St. Louis, MO, USA), 1% glutamine (BE17-605E, Cambrex, East Rutherford, NJ, USA) and 1% penicillin/streptomycin (P/S, 15140-122, Life Technology, Carlsbad, CA, USA). Cells were cultured to an 80% level of confluence in a humidified atmosphere of 5% CO_2_ at 37 °C. Transduction was performed on adherent cells at a concentration of 8000 cells/cm^2^ using green fluorescent protein (GFP), TMEM230-GFP (for TMEM230 upregulation) and shTMEM230-GFP (for downregulation) lentiviral vectors, respectively. The efficiency of transduction was verified by microscope observation. Cells were allowed to recover for two passages in adherent culture and used for immunofluorescence analysis and other assays.

### 4.3. Immunofluorescence Analysis

RA patient control and cells in which TMEM230 expression was modulated using the lentiviral approach (TMEM230mRNA and shTMEM230 cells) were fixed with 4% Paraformaldehyde (Sigma, St. Louis, MO, USA) in 1× PBS for 10 min at room temperature (RT). Cells were incubated with a blocking buffer of 5% normal goat serum in 1× PBS. The primary antibodies used were anti-phalloidin-TRITC conjugated (1:2000, Sigma, P1951) or anti-C20ORF30 (TMEM230, 1:1000, sc-85410, Santa Cruz, Dallas, TX, USA), all incubated for 2 h at RT. The cells were washed with 1× PBS and incubated with a secondary antibody of goat anti-rabbit Alexafluor-555 (1:500 dilution, A21429, Life Technologies, Carlsbad, CA, USA) for 1 h at RT.

### 4.4. Super-Resolution Microscopy

Super-resolution imaging data were collected by using a Stellaris 8 Falcon τ-STED microscope (Leica Microsystems, Mannheim, Germany) equipped with a supercontinuum pulsed (80 MHz) white light laser (WLL), a 775 nm donut-shaped STED (stimulated emission depletion) laser and an HC PL APO CS2 100×/1.40 oil immersion objective lens. The software LAS X (v4.5) was used for collecting images. Alexafluor594 was excited with both the WLL at 590 nm and the STED laser; its emission was collected in the range of 600–700 nm. The Hoechst, used for staining the nucleus, was excited at 440 nm, and its emission was collected in the range of 448–520 nm. For both fluorophores, we used hybrid detectors (Leica Microsystems) in counting mode with a pixel dwell-time of 0.72 µs and 6-line accumulation.

### 4.5. RNA-Seq Gene Expression Analysis

Differentially expressed genes associated with RA or OA were identified in patient datasets using *p*-values and log2 FC, as described in each table. Gene ontology and bi-ological pathways were assessed using the False Discovery Rate method (Benjamini) as previously described [6,7]. Functional enrichment analysis was performed using DAVID (version 6.8). Pathways identified were considered statistically if significant differences were associated with a *p*-values 0.05 or less.

## 5. Conclusions

Our sequencing analysis shows that both immune cells, macrophages and fibroblast cells were associated with altered expression of proteins previously associated with the endomembrane system of cells, including membrane-bound organelles such as the endoplasmic reticulum and mitochondria. How these genes contribute to autoimmunity and differences in immune response in RA compared to OA in the different cell types of the synovial tissue will need to be further investigated. An important observation in this study was that decrease or loss of expression of RNASET2 and TMEM230 were observed in all cell types, suggesting these genes are novel molecular markers in age-related human disorders due to their role in ER-dependent unfolded protein response in stress and aging and in normal ES-dependent tissue repair and regeneration. As an integral component of the ER, TMEM230 regulates glycan synthesis, glycan processing and glycolysis enzymes. Likely, the downregulation of TMEM230 observed in RA and concomitant misregulation of enzymes in glycolysis in RA may represent a paradigm for the study of other autoimmune disorders associated with aging.

## 6. Patent

Ileana Zucchi and Rolland Reinbold are recipients of EU Patent EP18707150.1, 6 September 2022 and US Patent US11566070B2, granted on 31 January 2023.

## Figures and Tables

**Figure 1 ijms-26-02412-f001:**
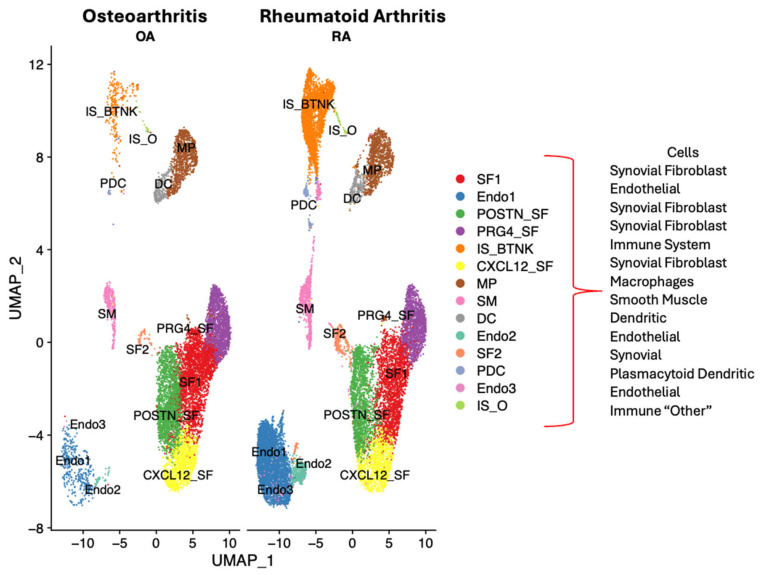
Osteoarthritis (OA) and rheumatoid arthritis (RA) synovial tissue transcriptomic map clustered by cell type. OA and RA high-resolution visualization of synovial tissue composition obtained by Uniform Manifold Approximation and Projection (UMAP) plot of synovial tissue cell type after integrating 3 OA (https://www.ncbi.nlm.nih.gov/geo/query/acc.cgi?acc=GSE152805, accessed on 30 December 2024) and 4 RA (https://www.ncbi.nlm.nih.gov/geo/query/acc.cgi?acc=GSE200815, accessed on 30 December 2024) datasets. Different clusters of cells were indicated by different colors according to the expression of representative specific markers as previously described in Angeli et al., 2025 [6] and Abeni et al., 2024 [7].

**Figure 2 ijms-26-02412-f002:**
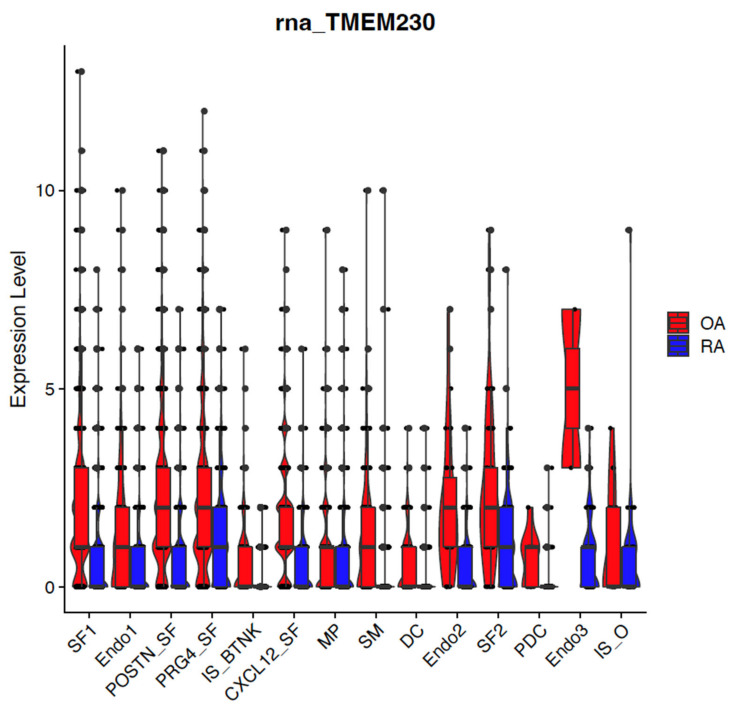
Box plot graphical visualization of the expression profile of the TMEM230 in OA and RA. The y-axis represents the expression level. *p*-value < 0.05 was calculated by the Kolmogorov Smirnov (K-S) test. TMEM230 expression is downregulated in all cell types from RA patients compared to OA. TMEM230 expression is very low or not detected in RA patient immune system cells (see DC, PDC and B, T and natural killer cells) compared to OA, suggesting that loss of TMEM230 promotes autoimmune features associated with RA patients. TMEM230 expression levels were previously identified downregulated in RA as described in Angeli et al., 2025 [6].

**Figure 3 ijms-26-02412-f003:**
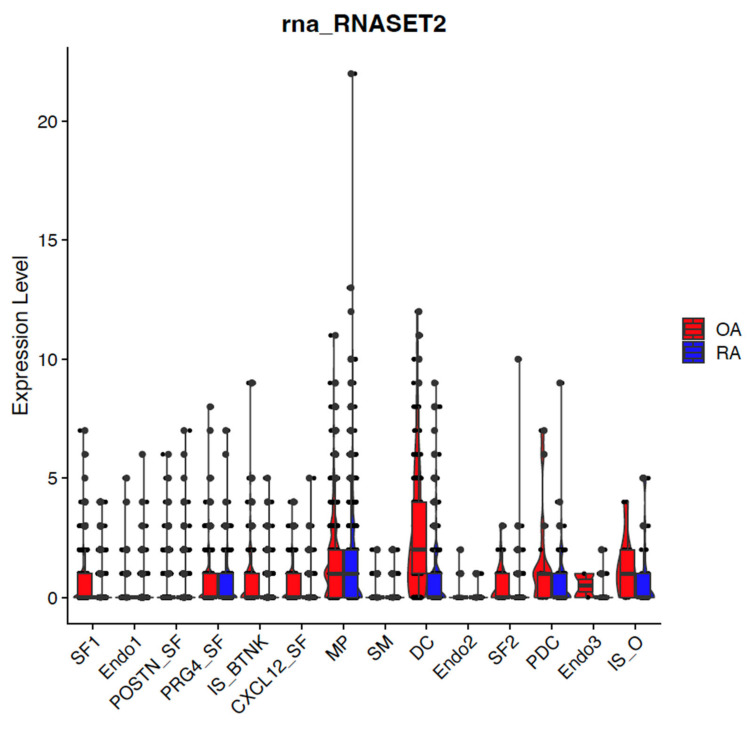
Box plot graphical visualization of the expression profile of the RNASET2 in OA (left panel, showing cell types with highest expression of RNASET2) and in a RA or OA comparative analysis (right panel). The y-axis represents the expression level. *p*-value < 0.05 was calculated by the Kolmogorov Smirnov (K-S) test. RNASET2 was previously identified as downregulated as described in Angeli et al., 2025 [6].

**Figure 4 ijms-26-02412-f004:**
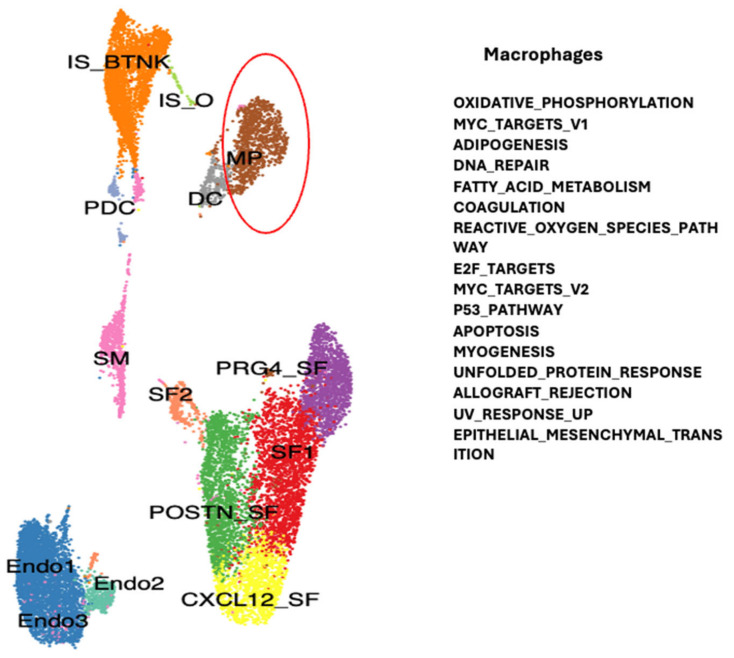
Gene and biological pathways differentially expressed between RA and OA patient synovial tissue macrophages may have a role in age-dependent autoimmunity (see Appendix A). Different clusters of cells were identified and indicated by different colors according to the expression of representative specific markers as described in Figure 1, Angeli et al., 2025 [6] and Abeni et al., 2024 [7]. Red circle indicates the macrophage cell containing cluster.

**Figure 5 ijms-26-02412-f005:**
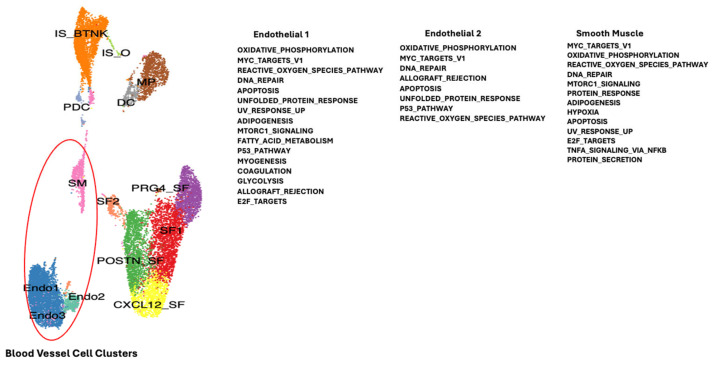
Gene and biological pathways differentially expressed between RA and OA patient synovial tissue associated blood vessel cells that may have a role in age-dependent autoimmunity (see Appendix A). Different colors are described in Figure 1, Angeli et al., 2025 [6] and Abeni et al., 2024 [7]. Red circle indicates blood vessel cells associated with synovial tissue.

**Figure 6 ijms-26-02412-f006:**
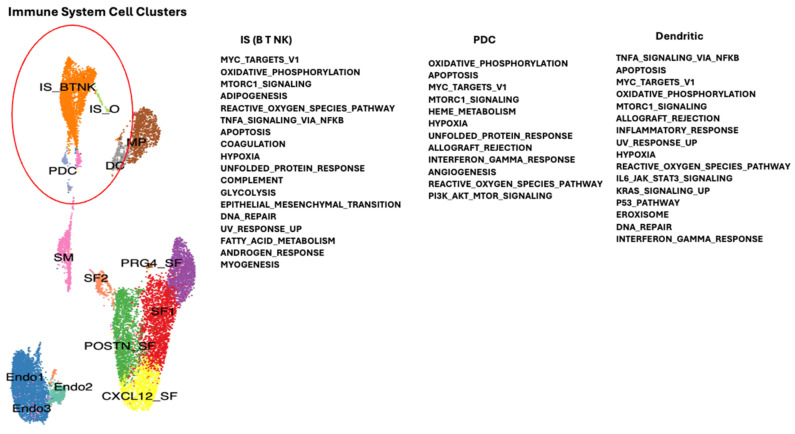
Gene and biological pathways differentially expressed between RA and OA patient synovial tissue-associated immune system cells that may have a role in age-dependent autoimmunity (see Appendix A). Different colors are according to Figure 1,Angeli et al., 2025 [6] and Abeni et al., 2024 [7]. Red circle indicates the different immune system cells associated with the synovial tissue.

**Figure 7 ijms-26-02412-f007:**
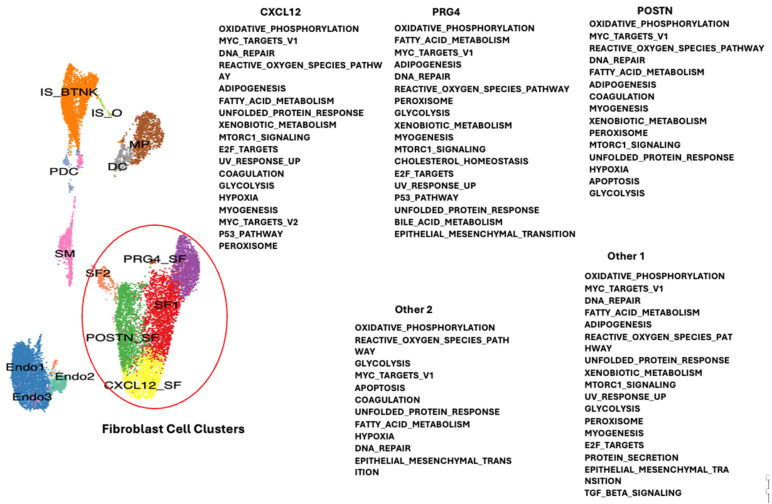
Gene and biological pathways differentially expressed between RA and OA patient synovial tissue fibroblast cells that may have a role in age-dependent autoimmunity (see Appendix A). Different colors are described in Figure 1 and Angeli et al., 2025 [6] and Abeni et al., 2024 [7]. Red circle indicates different types of synovial fibroblast cells of the synovial tissue.

**Figure 8 ijms-26-02412-f008:**
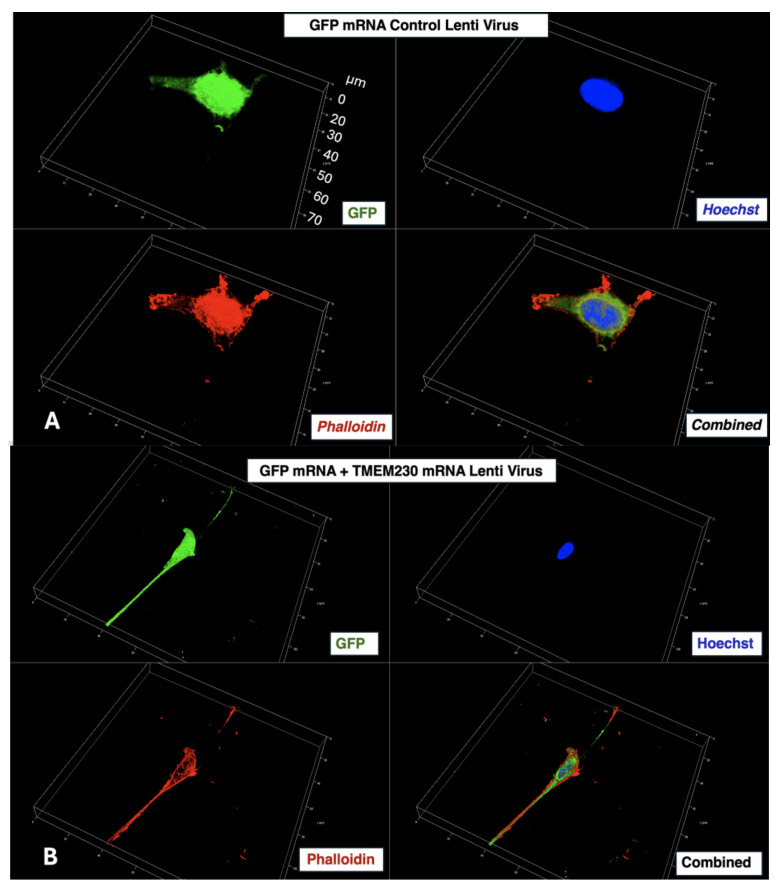
Super-resolution imaging of phalloidin and Hoechst in U87-MG cells of native levels of TMEM230 (green) (panel (**A**), labeled as GFP mRNA Lenti virus control) and in which TMEM230 expression was upregulated (panel (**B**)). Images were collected by using a Stellaris 8 Falcon τ-STED microscope. The Hoechst dye used for staining (blue) the nucleus was excited at 440 nm, and its emission was collected in the range of 448–520 nm. Alexafluor594 detection indicates expression of phalloidin (red) (590 nm and the STED laser, and emission was collected in the range of 600–700 nm). Antibody staining for phalloidin is performed as described in Cocola 2021.

**Figure 9 ijms-26-02412-f009:**
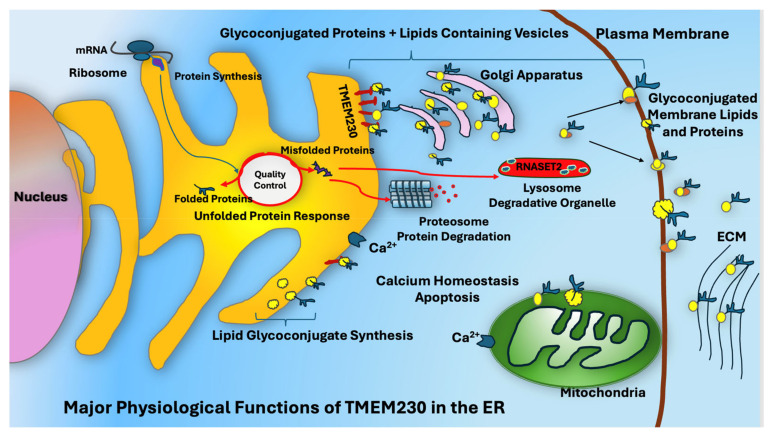
Major physiological functions of TMEM230 endoplasmic reticulum dependent regulation of glycosylation.

**Figure 10 ijms-26-02412-f010:**
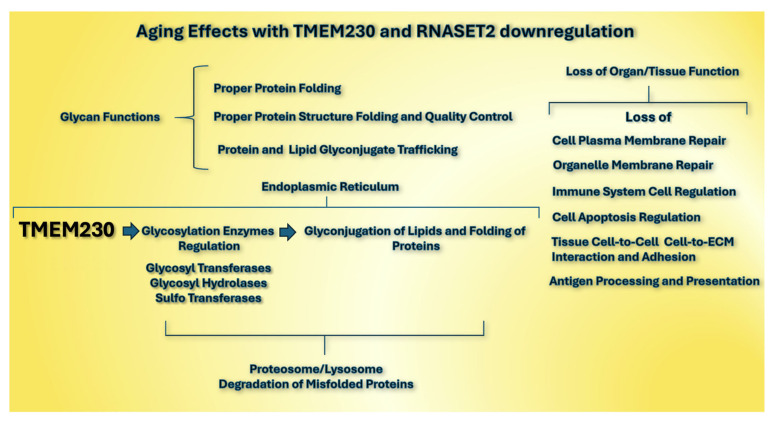
Possible pathological roles of aberrant regulation of glycan processing enzymes by TMEM230 in aging (see Appendix A and Figure 4, Figure 5, Figure 6 and Figure 7).

**Table 1 ijms-26-02412-t001:** Families of glycomic enzymes expressed in OA (red) or RA (blue) clusters in different cell types. The *p*-values associated with a fold change of specific enzyme mRNA are shown. Listed are the subfamilies of the glycomic enzyme genes belonging to the larger family Glycosyl Transferases (column GT), Glycosyl Hydrolase (column GH), Sulfo Transferase (column ST) and “Other” Glycan enzymes (column “other”) that are differentially expressed with a mRNA fold change with a *p*-value ≤ correct as authors want 0.05 in OA or RA clusters identified by single-cell RNA sequence analysis. Subfamilies were created using the classification of the Glyco-Enzyme Repository (http://glycoenzymes.ccrc.uga.edu) from the Carbohydrate-Active Enzymes (CAZymes) CAZy database. Glycomic enzyme expression was previously described in Angeli et al., 2025 [6].

		SF-1	ENDO1	POSTN_SF	PRG4_SF	IS_BTNK	CXCL12_SF	MP	SM	DC	ENDO2
Glycosyl Transferases	GT2			0.02				0.05			
	GT8				0.03						
	GT13	0.036								0.034	
	GT21	0.029	0.011		0.043		0.024		0.037		0.019
	GT29					0.024					
	GT31			0.05							
	GT39					0.042					
	GT41					0.022	0.046		0.046		

Glycosyl Hydrolases	GH13	0.036								0.03	
	GH20		0.024							0.01	
	GH22							0.0058		0.043	

	SULFO-TRANSFGERASES	0.0095					0.036	0.034		0.022	

	OTHER	0.0037		0.0095		0.0091			0.025		0.043

## Data Availability

Data is contained within the article and Appendix A or is available as described in Materials and Methods.

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
