# Peer review of "Glycosylation Regulation by TMEM230 in Aging and Autoimmunity"

_ijms, 2025, doi:10.3390/ijms26062412_

Round 1
Reviewer 1 Report
Comments and Suggestions for Authors
Dear Authors, thank you for the opportunity in reading your manuscript. I really do not have a real concern but I would like to know why did you choose rheumatoid arthritis as autoimmune disease and not other diesease in example: SLE which is the prototype of this kind of diseases. I would like to see if possible a schematic figure (maybe generated by BioRender) in which this immune and molecular pathophisiology could be better understood.
Thank you.
Author Response
Dear Reviewer
we appreciate very much the feedback, comments and suggestions for improving the manuscript and our research.
For the question why the author's choice rheumatoid arthritis is twofold, instead of for instance SLE. In the manuscript even though focus is on sequence analysis, we use and study primary patient samples. We do not have consent and approval to work with SLE patients, only rheumatoid arthritis. We assume that reviewers of our manuscript will ask for additional experiments using directly our patients with rheumatoid arthritis. Secondly, we have patients with osteoarthritis that do not have autoimmunity and find that using osteoarthritis patients that do not have systemic immunity issues is an appropriate control. We can obtain similar tissue (synovial tissue) from both patients. Which patients would be an ideal control for SLE is not clear for us and similar tissue from SLE and control patients may not be accessible for study.
We would be very pleased to add a schematic figure to help explain our hypothesis and the immune and pathophysiology of our study in our revision of the manuscript.
We thank the reviewer again for their very appreciated comments.
Reviewer 2 Report
Comments and Suggestions for Authors
TMEM230 (transmembrane protein 230), an ER protein, is known to be involved in secretory and recycling pathways. Mutations in TMEM230 or the abnormal expression may cause a defect in synaptic vesicle trafficking, autophagy, endosome recycling, Golgi secretion, and cell survival.
RNASET2, a lysosomal endonuclease, is known to have an association with autoimmune disorders (like rheumatoid arthritis, Crohn's disease, and Graves' disease), cancer, and inflammation.
The study by Gualtierotti and colleagues shows that TMEM230 was downregulated in all cell types from RA patients compared to control patients with osteoarthritis (OA). In the immune system cells of RA patients, the expression of TMEM230 and RNASET2 was very low, suggesting that the loss of TMEM230 and RNASET2 promotes autoimmune features associated with RA patients.
The authors discuss TMEM230’s role in PD across the manuscript as introductory/background information, but they forget to cite a study by Deng and Siddique’s group (PMID: 27270108). This was the study that established the role of TMEM23 in PD and showed mutations in TMEM230 impair synaptic vesicle trafficking.
The abstract is written quite waggly. The authors put a lot of introductory information in the abstract. When I read the abstract, it was difficult to understand where the main findings start. In the middle of the abstract, the authors discuss a few sentences about the findings, and again, the introductory information starts. I feel the abstract needs to be tightened up and should be concise.
Nothing new in the study. Just one more observation about the roles of TMEM230 and RNASET in rheumatoid arthritis. Authors reported similar observations in their previous study (PMID: 38960478).
The title of the manuscript should be changed. It looks like a review article’s title. The title should be descriptive.
Minor comments:
In Figure 5, the resolution of confocal images is not good. If authors can replace these images with images with better resolution, that would be great.
Author Response
The authors want to thank you for your very helpful comments and suggestions to improve the manuscript and our study.
TMEM230 (transmembrane protein 230), an ER protein, is known to be involved in secretory and recycling pathways. Mutations in TMEM230 or the abnormal expression may cause a defect in synaptic vesicle trafficking, autophagy, endosome recycling, Golgi secretion, and cell survival.
RNASET2, a lysosomal endonuclease, is known to have an association with autoimmune disorders (like rheumatoid arthritis, Crohn's disease, and Graves' disease), cancer, and inflammation.
The study by Gualtierotti and colleagues shows that TMEM230 was downregulated in all cell types from RA patients compared to control patients with osteoarthritis (OA). In the immune system cells of RA patients, the expression of TMEM230 and RNASET2 was very low, suggesting that the loss of TMEM230 and RNASET2 promotes autoimmune features associated with RA patients.
The authors discuss TMEM230’s role in PD across the manuscript as introductory/background information, but they forget to cite a study by Deng and Siddique’s group (PMID: 27270108). This was the study that established the role of TMEM23 in PD and showed mutations in TMEM230 impair synaptic vesicle trafficking.
We thank the reviewer for this important observation in reference to the role of TMEM230 in Parkinson's disease and also shown for other neurodegenerative diseases including Alzheimer's. There were several reasons why we did not directly address the excellent research of Deng and Siddique’s group (PMID: 27270108). One being our research is slightly different. The reviewer states that ". Our research is based on studying disease pathophysiology induced by aberrant expression of TMEM230. Our research from sequencing of TMEM230 shows that in autoimmunity and cancers we studied, that TMEM230 is not mutated but the diseases are do to epigenetic mechanisms that modulated incorrectly expression levels of TMEM230. How TMEM230 expression is modulated we do not yet know and is under investigation by our group. The study of Deng and Siddique’s group refers to mutations having a role in PD. The second reason why we did not address the studies of TMEM230 in PD or AD, is that there is controversy concerning the role of TMEM230 mutations neurodgeneration. We have had significant issues with other manuscript we submitted where other reviewers question whether mutations identified in other populations in other patients, such as in Italy. We will be very pleased to introduce this controversy in the discussion of our revised manuscript. We currently have no evidence that mutations in TMEM230 have a role in autoimmunity.
The abstract is written quite waggly. The authors put a lot of introductory information in the abstract. When I read the abstract, it was difficult to understand where the main findings start. In the middle of the abstract, the authors discuss a few sentences about the findings, and again, the introductory information starts. I feel the abstract needs to be tightened up and should be concise.
We appreciate the comments concerning the abstract and will endeavor to improve it and make it more concise and focused.
Nothing new in the study. Just one more observation about the roles of TMEM230 and RNASET in rheumatoid arthritis. Authors reported similar observations in their previous study (PMID: 38960478).
Although we have published TMEM230 and RNASET2 in rheumatoid arthritis, this study is unique because our study is the first were TMEM230 expression was manipulated using the lentiviral system. Our previous studies were bioinformatics studies with no patient in vitro cell work. In this study we provide direct proof of the role of TMEM230 in rheumatoid arthritis due to the fact we have patient and regulatory consent and approval for the first time to use patient derived samples. We apologize if this important information was not readily apparent. We will rewrite the text to make this more clear. Additionally, we have since when the paper was submitted performed in vitro analysis with TMEM230 gene expression modulation using the constitutive lentivral system. Previously TMEM230 expression modulation was performed in HUVEC, zebrafish, patient gliomas, fibroblast, macrophages from patients, but this is the first time that anyone has studied TMEM230 by directly modulating its expression in patient derived primary rheumatoid arthritis tissue cells. We propose to add additional in vitro experiments using RA patient cells to add further uniqueness and scientific merit to our study in the revised manuscript.
The title of the manuscript should be changed. It looks like a review article’s title. The title should be descriptive.
We will change the title. Thank you for this helpful comment.
Minor comments:
In Figure 5, the resolution of confocal images is not good. If authors can replace these images with images with better resolution, that would be great.
We had technical issues related to microscopy due to TMEM230 being an organelle and endoplasmic reticulum protein. We will re-perform imaging using alternative technical methods that may produce better images.
Round 2
Reviewer 2 Report
Comments and Suggestions for Authors
Please proofread the abstract part. Then, the MS will be ready for publication.